# Effects of Different Grafting Density of Amino Silane Coupling Agents on Thermomechanical Properties of Cross-Linked Epoxy Resin

**DOI:** 10.3390/polym12081662

**Published:** 2020-07-26

**Authors:** Dongyuan Du, Yujing Tang, Lu Yang, Chao Tang

**Affiliations:** College of Engineering and Technology, Southwest University, Chongqing 400715, China; ddy15520006078@163.com (D.D.); 18382427662@139.com (Y.T.); 15320343827@163.com (L.Y.)

**Keywords:** epoxy resin, silane coupling agent, grafting density, thermomechanical property

## Abstract

In order to study the influences of amino silane coupling agents with different grafting densities on the surface of nano silica on the thermomechanical properties of cross-linked epoxy resin, the molecular dynamics method was used to establish an amorphous model and calculate the mechanical properties, glass transition temperature, mean square displacement, hydrogen bond, binding energy, and radial distribution function of the composite models in this paper. The results are as follows: with the increase of the grafting density of an amino silane coupling agent on the surface of nano silica particles, the mechanical properties and glass transition temperature of epoxy resin showed a trend of increasing first and then decreasing. When the grafting ratio was 9%, the mechanical properties and glass transition temperature of the epoxy resin were the largest, and the glass transition temperature was increased by 41 K. At the same time, it was found that the higher the grafting ratio, the lower the chain movement ability, but the higher the binding energy. Besides, the binding energy between the nanoparticles of the grafted silane coupling agent and epoxy resin was negatively correlated with the temperature. By analyzing the hydrogen bond and radial distribution function, the results showed that the improvement of the grafted silane coupling agent on the surface of the nanoparticle to the thermomechanical properties of the epoxy resin was related to the OH···O and NH···O hydrogen bonds. The analysis results indicated that the proper grafting density should be selected based on the established model size, selected nanoparticle diameter, and epoxy resin materials in order to better improve the thermomechanical properties of the epoxy resin.

## 1. Introduction

As the main materials of the basin insulator, epoxy resin is very important for the safe and stable operation of high-voltage and ultra-high voltage equipment [1,2,3]. However, the solidified materials of epoxy resin have some inherent weaknesses such as low tenacity [4], low heat conduction [5], easy chapping, low insulation grade [3], and easy local discharge in a high-voltage field [6], which will result in a breakdown for the power equipment [7]. As the complexity and voltage grade of the operating environment grow, the requirements for epoxy resin become stricter. Research results showed that the epoxy resin doped with high-performance nanomaterials can improve the thermal performance, mechanical performance, and electric performance of epoxy resin effectively [8,9,10,11,12,13]. 

Nanoparticles have an excellent surface effect, small size effect, quantum size effect, macro quantum size effect, etc. [1]. Therefore, more and more domestic and foreign researchers have studied how to improve the performance of polymers by doping nanoparticles. However, the direct doping of nanoparticles is prone to agglomeration. Now, the silane coupling agent is frequently used to modify the surface of the nanoparticles and further reduce agglomeration. The reference [14] showed that the nanoparticles modified with an amino silane coupling agent can improve the glass transition temperature of epoxy resin and reduce its dielectric constant. Zhikun Wang [15] et al. found that the thermal stability of the epoxy resin can be improved and the glass transition temperature increases by 15 K after the KH550 was grafted on the surface of the SiO_2_ nanoparticles. Kung-Chin Chang [16] et al. found that the nanoparticles modified by the silane coupling agent can improve the thermomechanical performance of the epoxy resin and reduce the absorption of water. Kumarjyoti Roy [17] compared and studied the influences of 3-aminopropyltriethoxysilane, triethoxy(octyl)silane, and bis[3 (triethoxysilyl)propyl]tetrasulfide silane coupling agents with different functions on the rubber. Research results showed that the nanoparticles doped with bis[3-(triethoxysilyl)propyl]tetrasulfide can improve the mechanical effect and enhance the rubber hydrophobic property. Most of the current research focuses on polymers modified by nanoparticles grafting with silane coupling agents; there is less research on the grafting density of the amino silane coupling agents on the surface of the nanoparticles and its influences on the thermal performance of epoxy resin.

This paper studied the influences of N-(2-aminoethyl)-3-Aminopropyl trimethoxy silane coupling agents with different grafting densities on the surface of the nano silicon dioxide particles on the thermomechanical performance of the cross-linked epoxy resin and calculated thermomechanical parameters such as the dynamics performance, glass transition temperature, mean square displacement, hydrogen bond number, binding energy, and radial distribution function to select the optimal grafting density.

## 2. Materials and Methods

Bisphenol A epoxy resin (DGEBA) and 1,3 benzenediamine (BD) were selected respectively as the monomer and curing agent molecule of epoxy resin in this paper, and an epoxy resin composite model was established by Materials Studio (MS) software [18]. The C atom and N atom in the reaction between the epoxy resin monomer and curing agent molecule are respectively marked as R1 and R2. When the close-contact distance between R1 and R2 meets the preset distance, a cross-linked reaction will react to form a C–N cross-linked bond. For the schematic diagram of the cross-linked reaction, refer to Figure 1. Firstly, a 50×50×50 Å^3^ box in the Forcite module was established, and then DGEBA and BD molecules by a 2:1 ratio were placed into the box. Next, geometric optimization and molecule dynamics running were performed in order to make the established model reach the balance conformation, and finally, the cross-linked reaction was completed by using Perl language. COMPASS was selected as the force field for calculations in all dynamic simulations; Nose and Berendsen were used for temperature control and pressure control, respectively [18].

According to the literature [18], the N-(2-aminoethyl)-3-Aminopropyl trimethoxy silane coupling agent was selected to study the influences of the N-(2-aminoethyl)-3-Aminopropyl trimethoxy silane coupling agent with different grafting densities on the surface of the nano silicon dioxide particle on the thermomechanical performance of cross-linked epoxy resin in the paper. The diameter of the nano silicon dioxide particles is 6.6 Å [19]. Before the silane coupling agent is grafted on the surface of nano silica, it shall be firstly hydrogenated, which means that the oxygen atom is bonded with the H atom and the silicon atom is boned with the oxhydryl on the surface of nano SiO_2_ to meet saturated status. The silane coupling agent will hydrolyze to form an Si–O–Si bond on the surface of the nanoparticles [20]. Etienne [21] et al. found that the bond between the silane coupling agent and nanoparticle includes single-tooth status, double-tooth status, and three-tooth status by the experiments. The single-tooth status was selected in the paper. The schematic diagram of silane coupling agent grafted on the surface of nano silicon dioxide was shown in Figure 2. The pure epoxy resin, epoxy resin/SiO_2_, and epoxy resin doped with four grafting densities on the surface of the nanoparticles, which are represented as pure, 0, 3%, 6%, 9%, and 12% were established, respectively (0: the model with 0 graft silane coupling agent strips on SiO_2_; 3%: the model with 3 graft silane coupling agent strips on SiO_2_; 6%: the model with 6 graft silane coupling agent strips on SiO_2_; 9%: the model with 9 graft silane coupling agent strips on SiO_2_; 12%: the model with 12 graft silane coupling agent strips on SiO_2_).

## 3. Results and Discussions

### 3.1. Mechanical Performance

The static method [22] was used to calculate the elasticity constant in the paper, and the rigidity matrix of the composite material was calculated after balance operation. The second derivative of potential energy versus strain is shown as follows:(1)$$C_{ij}=\frac{1}{V}•\frac{\partial^{2}U}{\partial \varepsilon_{i}\partial \varepsilon_{j}}=\frac{\partial \sigma_{i}}{\partial \varepsilon_{j}}=\frac{\delta_{+}-\delta_{-}}{2\varepsilon_{j}}$$

∂ represents the stress, *U* represents the potential energy, “+” and “−” represent stretching and compression, and ε represents the strain. The Lame constant can be calculated as follows:(2)$$\left\{ \begin{array}{l} \lambda =\frac{1}{3}\left(C_{11}+C_{22}+C_{33}\right)-\frac{2}{3}\left(C_{44}+C_{55}+C_{66}\right)\\ \mu =\frac{1}{3}\left(C_{44}+C_{55}+C_{66}\right) \end{array}\right.$$

The young modulus *E*, volume modulus *K*, and shear modulus *G* could be calculated respectively as follows:(3)$$E=\frac{\mu (3\lambda +2\mu )}{\lambda +\mu }$$
(4)$$K=\lambda +\frac{2}{3}\mu$$
(5)G=μ

The young modulus, volume modulus, and shear modulus of different models were calculated, and the influence of temperature on each mechanical property is considered in Figure 3a–c respectively. From Figure 3, the mechanical properties will reduce when the temperature is rising. When the grafting density is less than 9%, with the growth of grafting density, the mechanical properties of the cross-linked epoxy resin model will increase. When the grafting density is 9%, the mechanical properties can reach the maximum. However, when the grafting density is 12%, the mechanical properties began to reduce. It shows grafting a silane coupling agent on the surface of the nano silicon particles can improve the mechanical properties of the epoxy resin, because grafting a silane coupling agent on the surface of the nanoparticles reduces agglomeration and make nano silicon particles fully react with the epoxy resin matrix. In different models, we should choose the appropriate grafting density on the surface of nanoparticles doped in the epoxy resin in order to obtain relatively good mechanical properties.

### 3.2. Glass Transition Temperature

The glass transition temperature (*Tg*) is an important characteristic parameter of the thermal performance of polymers, which could be calculated by the specific volume method [23,24,25], linear fitting between the free volume and temperature, or linear fitting between the energy and temperature [26,27]. The *Tg* values of different models were calculated within 300–650 K by using a specific volume method in this paper, which can be seen in Figure 4. The statistical glass transition temperature and experimental results in the references are shown in Table 1.

As shown in Figure 4, the glass transition temperature of the pure model is minimal. After the silane coupling agent was grafted on the surface of the nanoparticle, with the growth of grafting density, the glass transition temperature of the epoxy resin will first rise and then descend. When the grafting density is 9%, the glass transition temperature is the largest. Its increases by 41 K compared to the pure epoxy resin. When the grafting density is 12%, the glass transition temperature descends.

### 3.3. Mean Square Displacement

The mean square displacement (MSD) shows the atomic chain movement, mechanical performance, and thermal stability of polymers. The bigger the MSD value, the worse the mechanical performance and thermal stability of the polymers.

The MSD of pure, 0, 3%, 6%, 9%, and 12% model were calculated at 300 K in Figure 5. With the growth of the grafting density, the MSD is clearly starting to decline. When the grafting density is 12%, the MSD reaches its minimal value, and when the grafting density is 0, the MSD value is at its maximum. The research results indicated that the grafted silane coupling agent on the surface of the nanoparticles doped into epoxy resin can effectively reduce the chain movement of epoxy resin, make the structure of the epoxy resin more stable, and finally improve the mechanical performance and thermal performance of the epoxy resin. The research results show that the MSD of the pure model is lower than 3%, which is similar to outcome in the reference [12].

Figure 6 shows that the MSD value of the grafting density is 12% at 300 K–650 K temperature. With the growth of the temperature, the MSD will quickly rise. Therefore, the chain movement of the epoxy resin will be enhanced, which will lead to a decrease of the thermomechanical property. It further proves that the temperature can affect the structure and performance of the polymers. Figure 6 also shows that the chain movement of the epoxy resin will grow slowly with the temperature rising when the temperature is less than 450 K. When the temperature is over 450 K, the MSD value will quickly rise with the temperature rising. These results may be related to the glass transition temperature of the epoxy resin. It indicates that the temperature slowly affects the chain movement of the epoxy resin when the temperature is lower than the *Tg*. When the temperature is higher than the *Tg*, the temperature will intensify the chain movement of the epoxy resin and further make the thermomechanical performance of the epoxy resin reduce quickly.

### 3.4. Hydrogen Bond

The hydrogen bond is defined as the special intramolecular or intermolecular interaction. Reference [30] mentioned that the hydrogen bond grid among polymers will greatly affect the mechanical performance and aging resistance of the polymers. The hydrogen bond was defined by using the geometric rule [18] in the paper. The schematic diagram of the hydrogen bond is shown in Figure 7. The hydrogen bonds of NH···N and NH···O were mainly calculated in the paper. The relation between the temperature and hydrogen bond number at grafting density values of 3%, 6%, 9% and 12% is shown in Figure 8.

In Figure 8, with growth of the temperature, the number of hydrogen bonds will tend to reduce on the whole. The higher the temperature, the less hydrogen bonds there are. It indicates that the temperature destructs the hydrogen bond grid and further shows that polymers could not work under high temperature in the long term. The figure shows that the whole number of hydrogen bonds is greater than those of the models when the grafting density is 9%, and the number of hydrogen bonds of 9% increases by 11.4% when compared to the 3% grafting density model. So, the mechanical performance and glass transition temperature are better than those of other models when the grafting density is 9%. It also indicates that the different grafting densities on the surface of nano silica particles affects the epoxy resin performance due to the formed hydrogen bond between the epoxy resin and nanocomposite.

### 3.5. Interaction Energy

In order to study the interface adhesion energy between the cross-linked epoxy resin and nano silicon dioxide particle grafted with a silane coupling agent, the interaction energies of 3%, 6%, 9%, and 12% models were calculated. The interaction energy between the cross-linked epoxy resin and the nanocomposite could be calculated as follows:(6)$$E_{\mathrm{int}}=E_{total}-(E_{epoxy}+E_{Nonaparticles})$$

*E_total_* indicates all the potential energy of the cross-linked epoxy resin hybrid model, *E_epoxy_* indicates the potential energy of the epoxy resin, *E_nanomaterials_* indicates the potential energy of the grafting silane coupling agent on the surface of nano silicon dioxide particles, *E*_int_ indicates the interaction energy between the epoxy resin and the nano silicon dioxide particles’ grafted silane coupling agent.

The interaction energy between the cross-linked epoxy resin and nanocomposite models in 3%, 6%, 9%, and 12% models at different temperatures is shown in Figure 9. All the interaction energy between the cross-linked epoxy resin and nanocomposites is shown in Figure 9a, and the interaction energy between the cross-linked epoxy resin and nanocomposites in square angstrom is shown in Figure 9b. From Figure 9, we can find that the interaction energy of the four models is increasingly reducing with the rising of the temperature, and the interaction energy is negatively correlated with the temperature, which meets the following equation:(7)y=−A+Bx

*A* and *B* are constant, respectively. The result shows that the high temperature will destruct the interaction energy between the epoxy resin and the nano silicon dioxide particles, which is similar to the change trend before the calculated mechanical performance and the hydrogen bond at a different temperature. Figure 9a shows that the interaction energy will quickly grow with the growth of the grafting density. It indicates that the grafted silane coupling agent on the surface of nano silicon dioxide particles can significantly improve the interface adhesion energy between the epoxy resin and nano silicon dioxide particles. The purple broken line part of Figure 9a,b shows that the interaction energy quickly changes at 450 K temperature, which may have relations with the *Tg* of the polymers. It further indicates that the calculated *Tg* before is reasonable.

### 3.6. Radial Distribution Function

The radial distribution function (RDF) means the possibility of finding another particle at the radius *r* for any other atom [31,32]. It indicates the atom order problem and can also be explained by the ratio of the system’s regional density and mean density. The RDF could be calculated as follows:(8)$$g_{a-b}(r)=\frac{V\langle \sum_{i\neq j}\delta (r-\left|r_{Ai}-r_{Bi}\right|)\rangle }{N_{A}N_{B}-N_{AB}}4\pi r^{2}dr$$
where *i* and *j* are the *i*th atom and *j*th atom, respectively, while *N_AB_* indicates the number of common atoms between groups *A* and *B*.

The RDF values of all the atoms in the 3%, 6%, 9% and 12% models at 300 K temperature are shown in Figure 10. From Figure 10, we could find that the peak is 0 in the 0–0.9 Å due to the van der waals volume exclusion effect of atoms. The peak within 0.9–1.1 Å is from the chemical bond between hydrogen atoms and other atoms, the peak within 1.4–1.45 Å is from the C–N and C–O bonds, and the 1.75 Å peak is from the distance between the hydrogen atoms of methyl (-CH3) and methylene (-CH2-). The peaks in other molecules could be known as follows: the peak at R = 2.16 Å is from the H–C–C bond, and the peak at R = 2.44 Å is from the C–C–C bond [31,33]. When the RDF radial distance is over 4 Å, no peak occurs and the radial distance approximates to 1, because the established model system is amorphous [34].

By calculating the RDF of all the atoms of different grafting density models, it can be found that the RDF values of the four models are similar. Meanwhile, the RDF of the adjacent grafting densities model is slightly different, which may relate to the grafting number of the silane coupling agents between the adjacent grafting density models having a slight difference as well. In Figure 10, when R is within 0.9–2.5 Å, the RDF peak is slightly higher than that of the other three grafting models when the grafting density is 9%, while the peak is minimum when the grafting density is 3%, which is especially obvious within 0.9–1.1 Å, where the peak is mainly from the chemical bond in the model between the hydrogen atom and the other atom, including the hydrogen bond from the atoms. This result is consistent with that before the number of hydrogen bonds increased.

The RDF values of the C–H atom, N–H atom, and O–H atom for 4 different grafting density models are shown in Figure 11, Figure 12 and Figure 13. The results show that the RDF of the C–H atom has a similar variation trend with the RDF of all the atoms. When the grafting density is 9%, the RFD value of the highest peak is at its maximum. The two maximum peaks of the RDF of the N–H atom could increase with the increasing of the grafting density; when the grafting density is 12%, it has the highest peak. It may relate with the nitrogen atoms of the grafted silane coupling agent, which means that the higher the grafting density, the greater the number of nitrogen atoms. It will lead to the RDF of the N-H atom being more. However, when the grafting density is 9%, the RDF value between the O–H atom is maximum at the peak within 0.9–1 Å, while when the grafting density is 3%, the RDF value is at its minimum, but the RDF value at the peak within 2–2.5 Å will increase with the growth of the grafting density.

The above results show that the thermomechanical performance is relatively better when the grafting density is 9%, which may be related to the hydrogen bond between atoms in the nano composite model. By analyzing the RDF values of different grafting density models, the results show that such a hydrogen bond is related to the hydrogen bond type of OH···O or NH···O.

## 4. Conclusions

The influences of different grafting densities of SiO_2_ nanoparticles on the thermomechanical performance of the cross-linked epoxy resin were studied, and the mechanism was analyzed from the micro aspect by molecular dynamics. The conclusions are as follows:(1)The SiO_2_ nanoparticles grafted with silane coupling agents can effectively improve the thermomechanical performance of the epoxy resin. When the grafting density is 9%, it shows a better thermomechanical performance than the other models, and the glass transition temperature increases by 41 K. Duo to nanoparticles grafted with silane coupling agents improving nanoagglomeration, nanoparticles can fully react with the epoxy resin matrix to form a hydrogen bond grid, and such hydrogen bonds are mainly from OH···O and NH···O hydrogen bonds. However, when the grafting density of the silane coupling agents on the surface of the nanoparticles is too high, nanoparticles cannot effectively react with the epoxy resin, which will reduce the thermomechanical performance.(2)With the increase of grafting density of SiO_2_ nanoparticles, the chain movement ability of epoxy resin decreased, the interaction between SiO_2_ nanoparticles and epoxy resin was enhanced, and the interaction energy is negatively correlated to the temperature. Based on the different sizes of the established models, different nanoparticle diameters and epoxy resin materials are selected, so a proper grafting density should be selected to better improve the thermomechanical performance of epoxy resin.

## Figures and Tables

**Figure 1 polymers-12-01662-f001:**
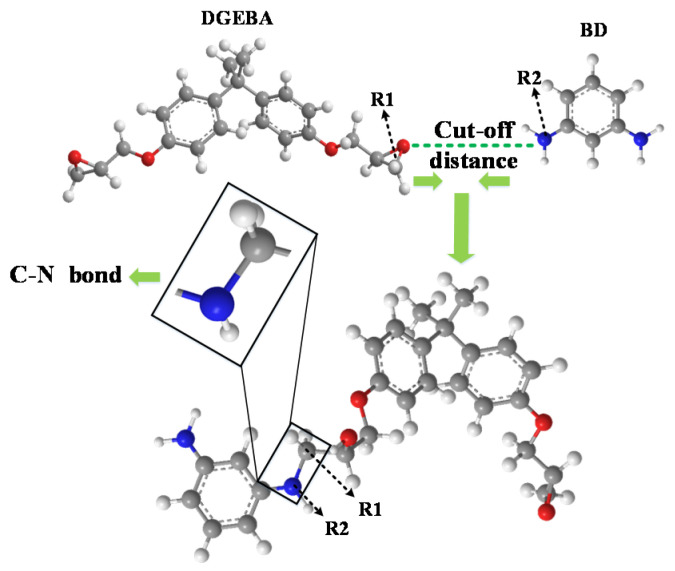
Schematic diagram of epoxy resin cross-linked reaction.

**Figure 2 polymers-12-01662-f002:**
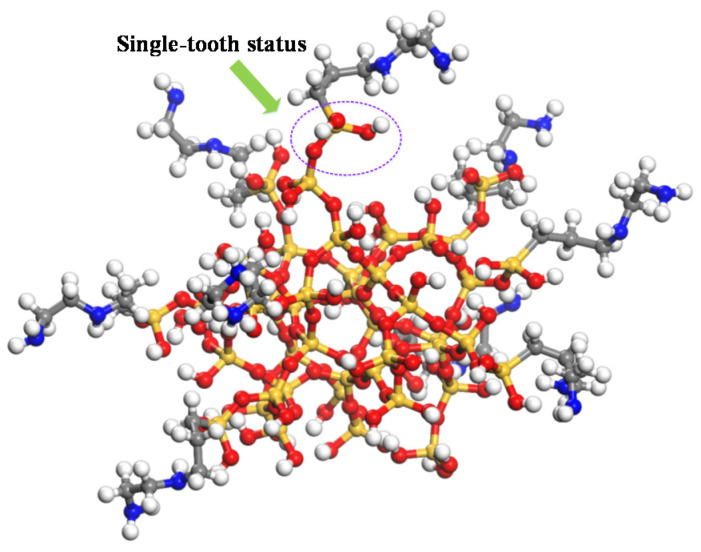
Schematic diagram of silane coupling agent grafted on the surface of nano silicon dioxide.

**Figure 3 polymers-12-01662-f003:**
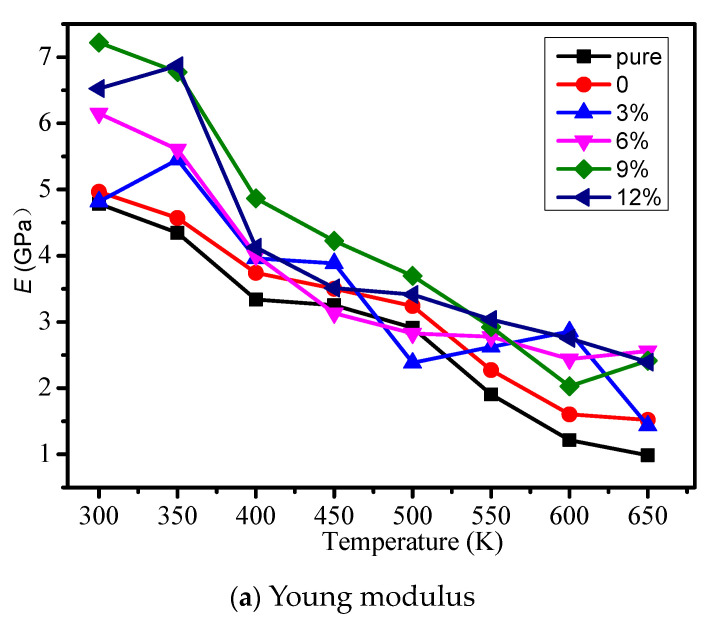
Mechanical property of different models. (**a**) Young modulus; (**b**) Volume modulus; (**c**) Shear modulus.

**Figure 4 polymers-12-01662-f004:**
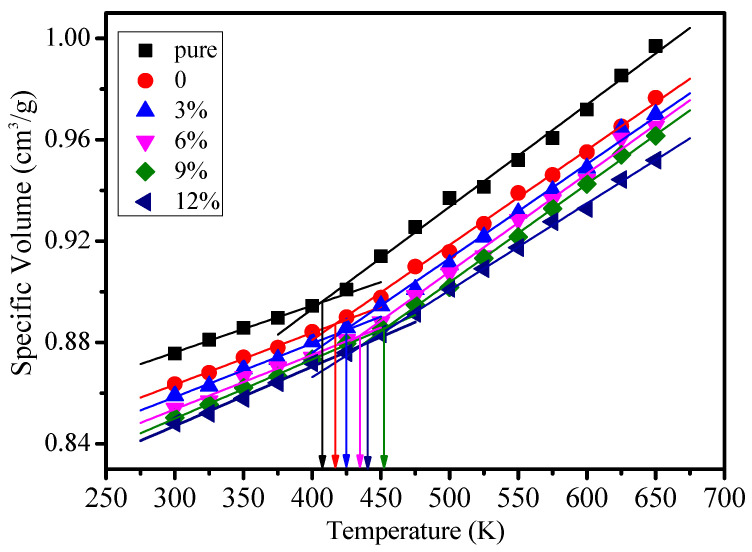
Specific volume value of different models at different temperatures.

**Figure 5 polymers-12-01662-f005:**
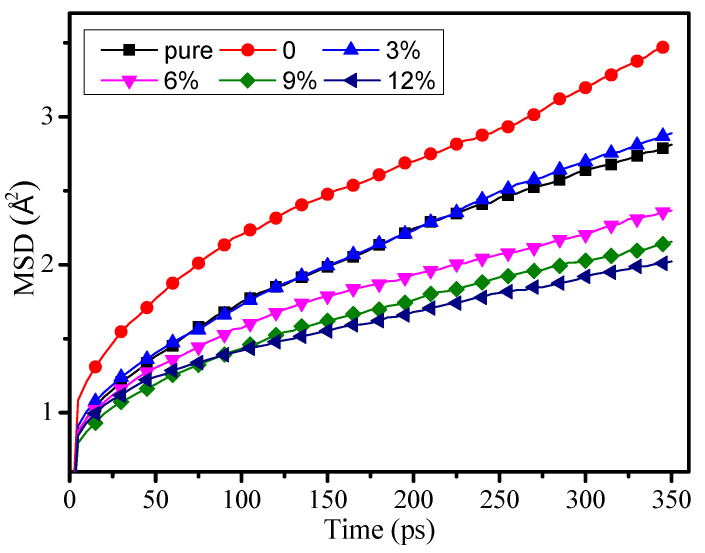
Mean square displacement of different models at 300 K.

**Figure 6 polymers-12-01662-f006:**
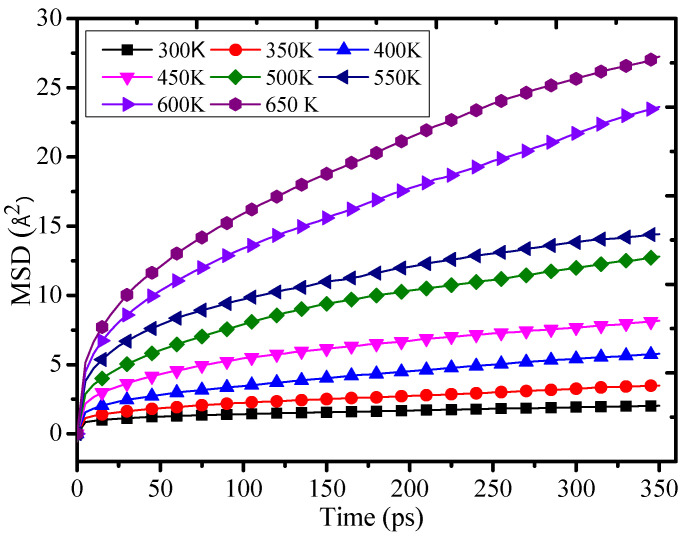
Mean square displacement of 12% at different temperatures.

**Figure 7 polymers-12-01662-f007:**
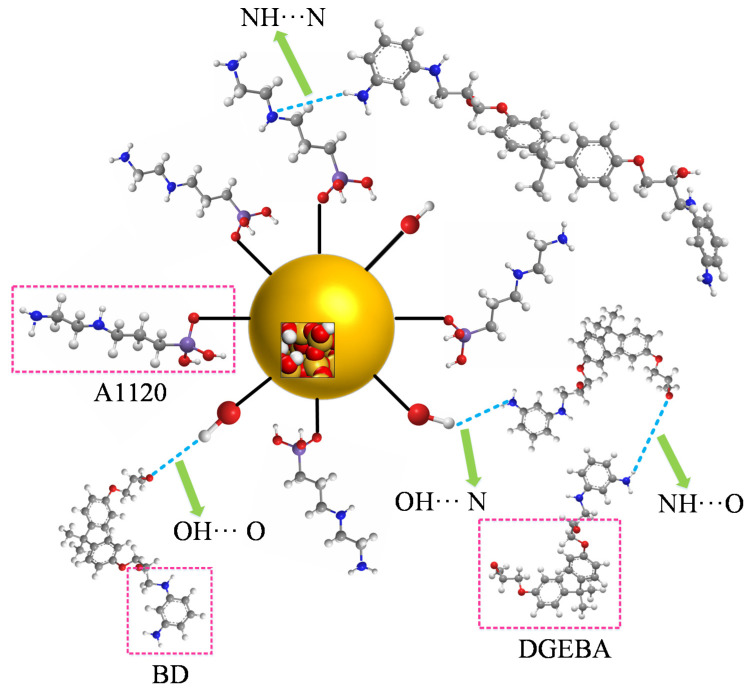
Schematic diagram of hydrogen bond.

**Figure 8 polymers-12-01662-f008:**
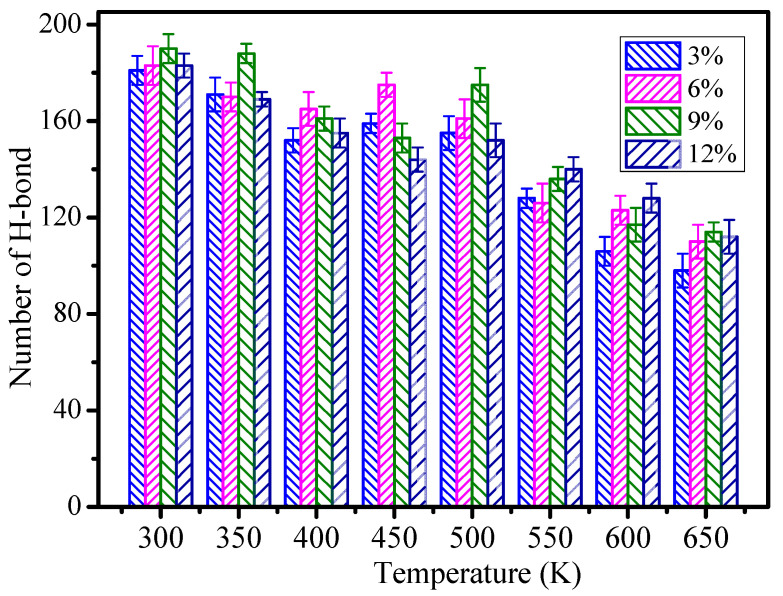
The number of hydrogen bonds of different grafting density models.

**Figure 9 polymers-12-01662-f009:**
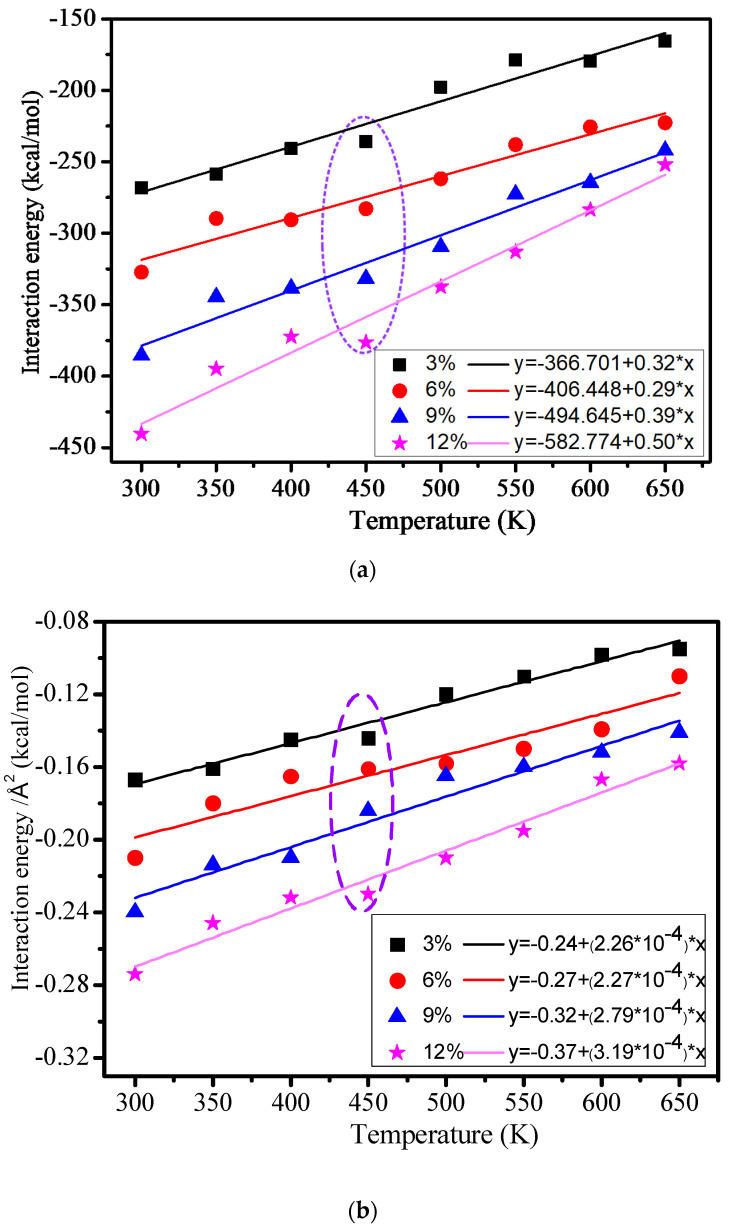
(**a**) All the interaction energy between the cross-linked epoxy resin and nanocomposite models; (**b**) Interaction energy between cross-linked epoxy resin and nanocomposite models in square angstrom.

**Figure 10 polymers-12-01662-f010:**
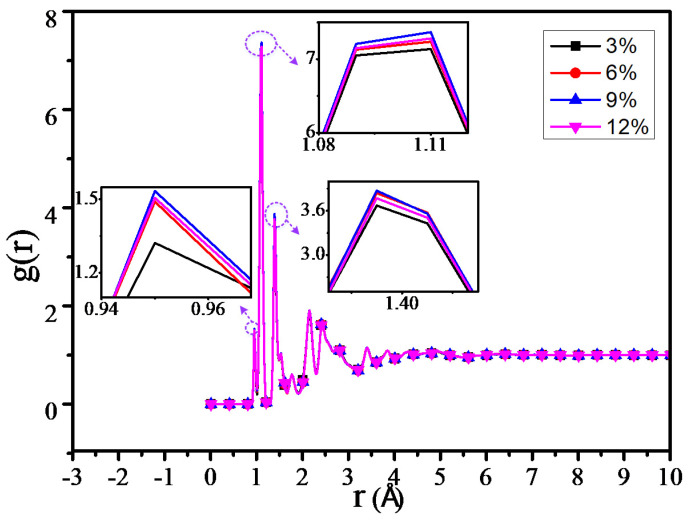
Radial distribution function (RDF) of all atoms of different models.

**Figure 11 polymers-12-01662-f011:**
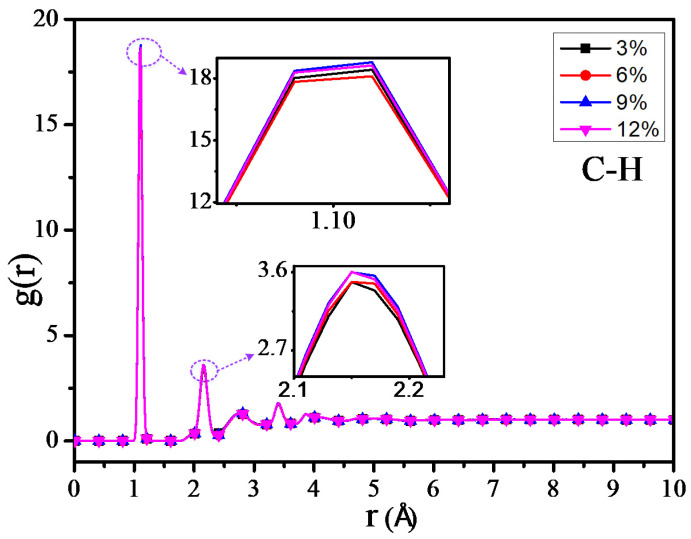
Radial distribution function (RDF) of C–H atoms of different models.

**Figure 12 polymers-12-01662-f012:**
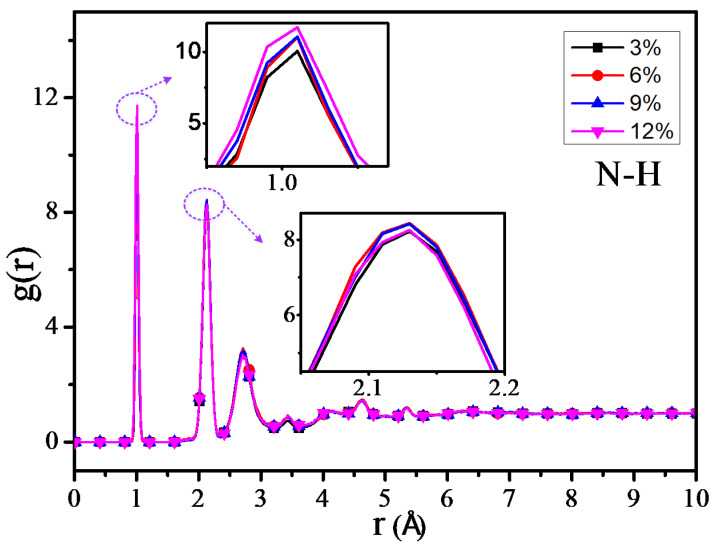
RDF of N–H atoms of different models.

**Figure 13 polymers-12-01662-f013:**
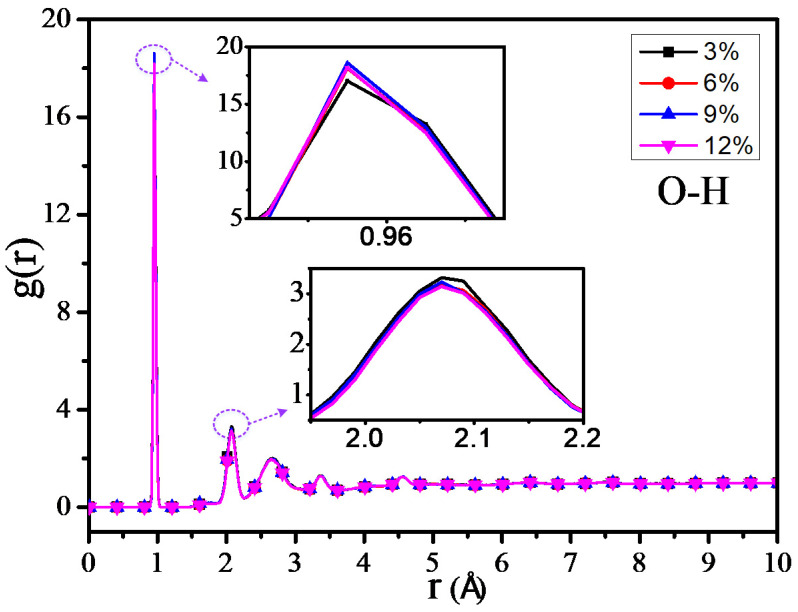
RDF of O–H atoms of different models.

**Table 1 polymers-12-01662-t001:** Glass transition temperature of different grafting models (K).

Model	Simulation Value	Experimental Value
pure	410	437 [28], 436 [29]
0	418	422 [28]
3%	425	
6%	433	
9%	451	
12%	437	

## Data Availability

All relevant data are included in the manuscript as Source Data; all other data are available from the corresponding authors upon reasonable request.

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
