# Peer review of "Effects of Different Grafting Density of Amino Silane Coupling Agents on Thermomechanical Properties of Cross-Linked Epoxy Resin"

_polymers, 2020, doi:10.3390/polym12081662_

Round 1

Reviewer 1 Report

The article by Dongyuan Du, Yujing Tang, Lu, Yang and Chao Tang is devoted to study the influences of amino silane (N-(2-aminoethyl)-3-Aminopropyl trimethoxy silane) coupling agent with different grafting densities on the surface of nano silicon dioxide particles on thermomechanical properties of cross-linked epoxy resin obtained on the base bisphenol A epoxy resin (DGEBA) and 1,3 benzenediamine (BD). The authors received a large amount of data that may be of interest to readers. However, the article requires some refinement in order to make it understandable to the reader. My comments are given below.

1) The authors use Materials Studio (MS) software but do not reference it anywhere. All appropriate references must be given.

2) It is not clear for me whether the degree of crosslinking of the polymer matrix was evaluated? What was the conversion rate of the matrix?

3) Also, I do not quite understand what was the weight fraction of the filler? How many nanoparticles have the authors placed in the matrix?

4) If I understand correctly, the coupling agent used can react with epoxy monomers. What was the proportion of reacted silane coupling agent at different NPs modification densities?

5) Very big doubts are caused for me the size of the used SiO2 nanoparticle. In case of the "NP" with diameter ~6.6A, the size of the such object is comparable to one unit cell of alpha-quartz or a molecule of silsesquioxane (POSS). So, in case of NP of 6.6A, the such "object" must contain only one unit cell of SiO2. Thus, the "molecular object" used by the authors has a very limited number of atoms where molecules of the surface modifier can be attached. Authors should clarify (check) the size of the used nanoparticles and it would be better to see - how many Si atoms were available (which may be bonded with the silane coupling agent) on the surface of the pure nanoparticle?

6) Page 3, Lines 89-91. The Authors write “The pure epoxy resin, epoxy resin/SiO2 and epoxy resin doped with four grafting densities on the surface of the nano particles, which are represented as pure, 0, 3%, 6%, 9% and 12% were established, respectively.”

It should be clarified how the authors understand the density of the modification. What, for example, does the number 3% mean? How many modifier molecules were sewn to the NP in this case? Since the number of atoms (which may be modifyed) is limited, it would be interesting to know what percentage of the total number of atoms was occupied in the case of a modification density of 3%? How much it was in the case of 6%? This question is related to the previous question about the size of the NPs used.

7) In the article, authors often use the designation for the prepared material “0, 3%, 6%, 9% and 12% model”, what does this mean? Samples with the different degrees modification of nanoparticles? A more understandable designation should be introduced. For example, on page 8 (lines 170 and 171), the authors write “The relation between the temperature and 170 hydrogen bond number in 3%, 6%, 9% and 12% can be known in the Fig. 8.” This is an example of an unsuccessful use of the above notation.

8) Page 9, lines 195-196. The authors write “The interaction energy between the cross-linked epoxy resin and nanocomposites models in 3%, 6%, 9% and 12% model at different temperature was shown in the Fig. 9 196”. It is not clear to me what is meant? Is it about the interaction of the matrix with NPs with varying degrees of surface modification, or is it something else?

9) In fig. 11-13 the meaning of the sidebars is not entirely clear. What do the authors want to demonstrate with them? I think that they are superfluous. They demonstrate the discreteness of RDF behavior, which is associated with the choice of step when plotting.

Reviewer 2 Report

The manuscript discusses the effect of different grafting density of amino silane coupling agents on the thermomechanical properties of cross-linked epoxy resin. The study is very useful and would be beneficial for the readers. Therefore, I recommend a minor revision of the manuscript.
The authors need to address these specific applications.
It is suggested to some details on the methodology for the simulation, even though it is obtained from the previous studies.
What is the degree of cross-linking? Was it fully cross-linked, 100%? Or less than that.
In section 3.2, the glass transition temperature in this paper is compared with that in the reference. Is there any comparison of the mechanical properties?
Overall the language of the paper should be improved.

Round 2

Reviewer 1 Report

The authors fully answered my questions and I have no new comments.